# Reimagining Healthcare: Human–Animal Bond Support as a Primary, Secondary, and Tertiary Public Health Intervention

**DOI:** 10.3390/ijerph20075272

**Published:** 2023-03-27

**Authors:** Janet Hoy-Gerlach, Lisa Townsend

**Affiliations:** 1OneHealth People-Animal Wellness Services (OHPAWS), Toledo, OH 43606, USA; 2Center for Human-Animal Interaction (CHAI), Virginia Commonwealth University, Richmond, VA 23284, USA; 3Departments of Pediatrics and Psychiatry, Virginia Commonwealth University, Richmond, VA 23284, USA; 4Children’s Hospital of Richmond, Richmond, VA 23230, USA

**Keywords:** one health, Ottawa Charter, public health, human–animal bond, human–animal support services, salutogenic, wellness

## Abstract

The emergence of human–animal support services (HASS)—services provided to help keep people and their companion animals together—in the United States has been driven by two global public health crises. Despite such impetuses and an increasing recognition of One Health approaches, HASS are generally not recognized as public health interventions. The Ottawa Charter, defining health as well-being and resources for living and calling for cross-sector action to advance such, provides a clear rationale for locating HASS within a public health framework. Drawing from Ottawa Charter tenets and using the United States as a case study, we: (1) recognize and explicate HASS as public health resources for human and animal well-being and (2) delineate examples of HASS within the three-tiered public health intervention framework. HASS examples situated in the three-tier framework reveal a public health continuum for symbiotic well-being and health. Humans and their respective companion animals may need different levels of intervention to optimize mutual well-being. Tenets of the Ottawa Charter provide a clear rationale for recognizing and promoting HASS as One Health public health interventions; doing so enables cross-sector leveraging of resources and offers a symbiotic strategy for human and animal well-being.

## 1. Introduction 

While efforts to promote health and reduce disease for humans and for non-human animals (henceforth referred to as animals for ease of reference) are often siloed by sector, the importance of the human–animal bond for mutual well-being is being increasingly recognized across human and animal health and welfare organizations across the globe. Fine and Beck [1] review the history of the human–animal bond from ancient civilizations forward, noting how humans’ relationships with animals evolved from using them to accomplish tasks to living with them as companions. A growing body of research indicates that companion and therapy animals mitigate deleterious stress responses and facilitate social interactions [2] and that some animals, particularly dogs, may promote physical exercise and subsequent cardiovascular health [3]. Focus group samples of dog owners explicate people’s subjective perceptions that their dogs confer physical, psychological, and social benefits [4]. Meta-analytic work extends these findings to animal-assisted therapy, suggesting that adjunctive treatment involving trained therapy animals is linked with improvements in medical, behavioral, and emotional well-being [5]. While numerous studies suggest that a bond with animals brings positive outcomes, findings must be interpreted cautiously. Amiot and Bastian [6] consider the animal side of the human–animal bond and whether relationships with humans lead to benefits for animals. This is particularly important given that animals are frequently employed as tools, served as food, and do not have the same agency that people do. Rodriguez, Herzog, & Gee [7] discuss the myriad of methodological explanations for the varied findings in human–animal interaction research outcomes. The questions “when” and “for whom” are vital considerations in interpreting research on the relationship between humans and animals, as is the degree of temperamental fit between individual humans and animals [6]. 

The critical necessity of environmental well-being is likewise being recognized and increasingly addressed in concert with human and animal welfare, rather than as an issue separate from human and animal well-being. The term One Health has emerged to give name to this interdependence between human, animal, and environmental well-being; simply put, a One Health approach entails focusing on the connections between human, animal, and environmental well-being in order to maximize positive outcomes within and across these areas [8]. 

In order to operationalize the potential of One Health approaches into actionable change for increased well-being, collaboration across a range of societal sectors is necessary. The advent of human–animal support services (HASS) in the United States to keep people with AIDS and their companion animals together as long as possible during the 1980s AIDS epidemic and the renaissance of HASS in the States following the COVID-19 pandemic serve as One Health examples for which cross-sector collaboration from human and animal health/welfare organizations has been critical in achieving positive outcomes for vulnerable and interdependent people and animals. The Ottawa Charter [9] explicitly calls for such cross-sector collaboration and expansively explicates health as going beyond disease to encompass well-being and the resources and infrastructures that support such. The purpose of this paper, drawing from tenets of the Ottawa Charter, is to: (1) recognize and explicate services that support human-companion animal relationships, e.g., HASS, as public health resources for human and animal well-being and (2) to delineate examples of human–animal support services (HASS) within the three-tiered public health intervention framework. The framework we introduce focuses specifically on humans and companion animals who live together.

## 2. Understanding Health: More Than Absence of Disease

In considering HASS as a One-Health-focused public health support, it is useful to begin by considering what is meant by “health”. Defining “health” has proven to be an elusive task, involving contributions from physicians, spiritual healers, philosophers, and scientists over centuries. Until the 19th century, health in Western cultures was viewed primarily through a spiritual lens, in which infections and disease were forms of curses [10]. As understanding of germs and bacteria emerged, a mechanistic understanding of disease emerged in which ill health was understood as physical body malfunctions that could be targeted with curative attempts. This mechanistic view, along with the work of Descartes [11], which posited mind–body dualism, became what is known as the biomedical model. The biomedical model entails specific notions of health, illness, and disease and became the dominant medical paradigm in Western culture until the 20th century; many would argue that it is still the prevailing model [10]. Within the biomedical model, health is understood simply as an absence of disease; illness entails a biological disease state that is explicit to the physical body and not changed by context [10]. Critiquing such a reductionist approach, Engel proposed the biopsychosocial model, in which psychological and social factors were integrated into a biological understanding of health [12]. 

While the biomedical model that health is the absence of disease or disability has predominated clinical and scientific approaches and continues to do so, numerous theoretical models have emerged that attempt to operationalize what being healthy means [13]. The understanding of health as well-being rather than absence of disease was articulated in Antonovsky’s Salutogenic Model [14]. According to the Salutogenic Model developed by Antonovsky [14,15], the human conditions of health and illness can be understood as existing on one continuum that an individual moves along, rather than as discrete and dichotomous states. Movement along the continuum of health and illness is construed as affected by an individual’s personal, physical, social, and environmental resources [14,15]. The Salutogenic Model inclusion of focus on factors supporting increased health rather than solely focusing on factors related to prevention and treatment of disease entails a paradigm shift in public health and health/mental health care [16]. Simply put, salutogenic approaches focus on increasing and supporting what keeps people healthy rather than on what makes people ill. Additional models have emerged that focus on well-being; the Wellness Model focuses on health promotion strategies and progression toward higher levels of functioning, and the Environmental Model frames health as entailing harmony with one’s physical and social surroundings [13].

Such understandings of health as encompassing far more than mechanistic interventions on individuals to address physical disease have likewise emerged as a driving force in international health policy. As explicated by the Constitution of the World Health Organization (WHO), health is defined as “…a state of complete physical, mental, and social well-being, and not merely the absence of disease or infirmity” [17] (p. 1). At the First International Conference on Health Promotion, an international agreement known as The Ottawa Charter for Health Promotion of 1986 was developed and called for reframing health as a “positive concept emphasizing social and personal resources” that supports well-being. The Ottawa Charter has been referred to as the internationally recognized “central document of health promotion” [16] (p. 190). An important tenet of the Charter is that health “is a resource for life” and not the goal of living [9]. 

The essential message from the First International Conference on Health Promotion was a call to mobilize resources at the individual, family, community, and sociopolitical levels to advocate for health as well-being. This includes enabling individuals’ agency to modify social determinants of health and promote social and personal resources to enhance well-being. The Ottawa charter goes beyond a physical disease/absence of disease conceptualization of health to include social and emotional wellness. Areas for priority action identified in the Ottawa Charter include building healthy public policy inclusive of health-promoting factors; creating supportive environments, including protection and conservation of natural resources; strengthening community actions; developing personal skills; reorienting health services beyond a focus on clinical and curative services; and focusing on care, holism, and ecology [9]. 

Rather than focusing on traditional public health interventions for populations facing specific health related issues or risks, the Ottawa Charter calls for mediation beyond the health sector, e.g., for coordinated efforts that protect and promote health through reconciling interests across a wide range of societal sectors [9]. Health-promoting changes in lifestyles and environmental conditions can create cross-sector conflicts related to access/use/distribution of resources and constraints on sector practices; proactive recognition, mediation, and reconciliation of such is a crucial component of the Ottawa Charter that is necessary for sustainable change [9]. The mediation called for in the Ottawa Charter is exemplified in many One Health strategies, as a One Health approach necessitates working across different sectors.

## 3. HASS: A One Health Approach

Cross-sector One Health approaches that support human and animal well-being can be traced from origins in ancient healing practices through an integrated veterinary and human public health system in 11th–13th century China, to modern day comparative medicine approaches [18]. However, the primary focus of One Health efforts has in recent history been on prevention of zoonotic diseases and preservation of human health. A One Health paradigm emerged in 2003–2004 following the appearance of severe acute respiratory diseases (SARS and H5N1) and recognition of the interconnection between humans, animals, and their environments [19]. Its primary focus has been the prevention of zoonotic illnesses and preservation of human health. For example, the World Health Organization (WHO) adopted a One Health approach to investigate the origins of the SARS-CoV-2 virus, including potential initial animal hosts, intermediate animal reservoirs, subsequent viral contamination of food products, and finally, human-to-human contagion [20]. The European Union One Health Zoonosis report summarizes data on transmission of illnesses to humans via consumption of animals or animal products [21].

While human well-being is of course important, a One Health focus that fails to recognize the interdigitation of human and animal well-being risks excluding factors that enhance biodiversity, a necessary element for survival of all species [22]. Zinsstag et al. [18] call for a “health in social-ecological systems” One Health view that prioritizes the health of both humans and animals and acknowledges the bidirectional influence of human/animal health on their social and ecological environments. A standout exemplar of this unified approach is Chandler et al.’s [23] review outlining common predisposing factors for obesity in people and their companion animals and proposing a One Health approach to preventive interventions for both. Fine [24] expands these ideas, suggesting that an integrated One Health approach must encompass social justice, animal welfare, and environmental protection alongside human health. 

HASS are an example of a One Health approach that integratively advances human and animal well-being and social justice. HASS are an international phenomenon, and HASS innovations are flourishing in nations across the world. To focus scope and analysis, HASS are examined within this manuscript in the context of the United States as a single-country case example. The use of the term human–animal support services (HASS) emerged in the United States during the early 1980s AIDS epidemic in California and was also referred to as the San Francisco Model; as people were debilitated with AIDS, a groundswell of grassroots volunteer efforts emerged to help keep people with AIDS with their animal companions for as long as possible and help ensure pets who lost their people to AIDS were cared for [25]. HASS entail the provision of needed assistance—such as pet food banks, accessible/affordable veterinary care, and pet housing deposit assistance—aimed at keeping vulnerable people and their companion animals together, typically for mutual benefit of the person and animal [25,26]. These grassroots efforts evolved into a formal organization in 1987—Pets Are Wonderful Support (PAWS)—through a collaboration of AIDS activists, people living with AIDS-related disabilities, veterinary and animal welfare professionals and students, and human health and social service professionals [25,27]. PAWS provided a range of services, such as pet food assistance (pet food banks and home delivery), financial and logistic help accessing veterinary services, and in-home pet care assistance (dog walking, emergency fostering, cat care, etc.) and eventually created a community tool kit to assist other localities across the country in developing such services. PAWS merged with the Shanti Project in 2015 and continues to provide HASS development and leadership across the United States [27].

Another mass health crisis—the COVID-19 pandemic—prompted a rapid expansion of HASS in the United States. Unlike the launching of HASS services in the 1980s which originated in the United States through the efforts of human health/AIDS activists, this 2020 expansion of HASS was driven from the animal welfare sector. Through a national call for collaborative efforts launched by American Pets Alive in March 2020 [26], collaborating animal welfare partners took the following actions to help keep people and animal companions together: HASS models were rapidly created and piloted in June 2020 in 33 cities across the United States, a HASS website was launched in 2020, the Keeping Families Together Eviction Response Toolkit was released in 2021, and specific HASS tools were subsequently developed and released online [26]. Partner animal welfare organizations in the United States that are implementing HASS in collaboration with other sectors in their communities can be viewed at https://www.humananimalsupportservices.org/partners/ (accessed on 30 January 2023).

## 4. Understanding HASS as Public Health Interventions 

Both the human-health-sector-driven origination of HASS in the 1980s in the United States during the AIDS epidemic and the animal-welfare-sector-driven expansion of HASS in the United States beginning in 2020 with the advent of the COVID-19 pandemic can be understood as needed public health responses to severe public health crises. HASS services can promote human and animal well-being, support people and animals at risk of adverse outcomes, and/or remediate negative conditions being experienced by people and/or their animal companions. HASS can thus be located within the three-tier framework developed in the public health sector—the tiers being primary, secondary, and tertiary prevention—to distinguish points of intervention at which effective strategies may be used to prevent or reduce the impact of disability, disease, and injury [28]. 

Primary prevention refers to interventions taken to prevent the occurrence of a disability, disease, or injury by reducing associated behavioral and environmental risk factors; the interventions are taken with the general population rather than a specific subpopulation identified as at particular risk for a given health problem [29]. Helfer et al. [30] (p. 850) further distinguish between primordial prevention and primary prevention, as follows: 

The goals of disease prevention are to avoid the development of risk factors, which have not yet occurred (primordial prevention), and to reduce the risk factors, occurrence and spread of disease (primary prevention). Additionally, early detection is intended to reduce the incidence of diseases (secondary prevention).

Within the Institute of Medicine Report [31], prevention is described as interventions that enable delaying or avoiding health issues and are divided into three types—universal, selective, and indicated—to reduce new cases. Universal prevention is focused on an entire population, whereas secondary selective prevention is aimed for those identified as at risk of developing a disorder and indicated prevention is done with those who have developed the disorder [31]; tertiary prevention is often referred to as “treatment” of health/mental health issues within everyday health and mental health practice. While in practice, health prevention and health promotion services are often conflated, health promotion is a term in public health that is conceptually differentiated from prevention in that:

“...while health promotion aims at improving/promoting health and resources (salutogenesis), prevention focuses on avoiding disease and its associated risk factors. In practice, however, health promotion and prevention often have the same goal, which is to improve or maintain the health and mental health competencies of the population. Consequently, they have been combined… [h]ealth promotion focuses on strengthening the personal resources of individuals and groups as well as improving the resources present in the social environment.” [30] (p. 851).

To better understand and develop knowledge on how proactive strategies for supporting the mutual benefits of the human–animal bond as both remediating and salutogenic health factors, the existing tiering for public health interventions can be used to situate One Health/One Welfare interventions for humans and animals. While “One Health” focuses on interconnections between human, animals, and the environment for the purpose of disease prevention, One Welfare focuses on human/animal/environment interconnection for enhancing the well-being of each [32]. In light of the foregoing discussion regarding the mutuality of health and well-being outcomes for humans and companion animals, and the import of cross-sector efforts to mobilize and maintain well-being as articulated in the Ottawa Charter, we propose a One Health/One Welfare (OHOW) public health model within which HASS services can address a spectrum of needs for humans and animals who live together. As an accompaniment, we provide several examples of HASS in each of the three public health intervention tiers (Table 1).

At its most basic, and in keeping with the Ottawa Charter, HASS within the OHOW model offer basic, primary, preventive practices to promote health and well-being for humans and companion animals. These include generally accepted health practices for humans, such as healthy eating, sufficient sleep, and time for hobbies and relaxation. For humans, an array of research findings describing biological, psychological, social, and emotional benefits related to animal companionship and the human–animal bond are emerging [41]. For instance, based on the empirical evidence pointing to the protective cardiovascular health benefits of living with companion animals, the American Heart Association recently launched the national public health campaign “Healthy Bond for Life” to raise awareness of such benefits [42]. The animal side for companion animals includes preventive veterinary visits; high quality, nutritious pet food; positive behavioral training and socialization with compeers and people; and time for play and enrichment. These are typically conveyed in Western cultures via the companion animal belonging to and living with a particular individual(s). HASS that promote people and companion animals forming bonds and living and spending time together, with the associated benefits afforded for mutual well-being, are thus primary health promotion. As illustrated in Table 1, initiatives that focus on making cities more welcoming of people being with their companion animals in public spaces can be broadly understood as HASS primary health promotion intervention.

The secondary level of OHOW is designed for people or animals at risk of negative health or well-being outcomes. On the human side, this could mean targeted, supportive human–animal interaction interventions for a youth who has experienced school bullying, or reading-to-dogs enrichment services for a child who is a reluctant reader. Selective animal interventions could encompass specialized training for a fearful shelter animal or an activity-based program for dogs prone to developing obesity. It is worth noting that human–animal bonds can pose stressors as well as strengths for people and animals; HASS interventions that ameliorate risk factors associated with human–animal bond stressors can be thus located as secondary interventions. For instance, within Table 1, HASS in the form of pet deposit fee assistance offers an example of alleviating housing instability for people and animals at risk of becoming unhoused.

OHOW’s final and tertiary level involves intervening to lessen the effects of a negative outcome that has occurred or cure a disorder for people and/or animals. Obesity and its downstream consequences represent one example of an illness that affects both people and animals. In this case, both humans and animals might receive treatment for obesity-related illnesses, such as diabetes, and could participate in weight reduction programs, such as dog-walking, together. Within Table 1, a program that places animals in shelters with people struggling with mental illness symptoms and isolation provides an example of a tertiary intervention; the animal without a home or human caregiver is adopted, and a person experiencing mental health symptoms and isolation receives the ameliorating effect of animal companionship on these issues. 

The examples above illustrate the simplistic case of people and animals residing on the same points of intervention—primary, secondary, or tertiary—on the OHOW continuum. This makes sense given that people and their companion animals will necessarily be exposed to similar environments—this could be symbiotic in effect or mutually deleterious. It may also be the case that people and their companion animals may require different levels of intervention (Table 1) while living in the same environment. 

HASS can be understood as a mechanism for equitable access to the resources needed to support the human–animal bond and the mutual benefits it affords to people and animals as well as to prevent mutually harmful severing of the bond during crises. Environmental and societal contributions to poor health are consistently recognized as important targets for public policy intervention [43]. Apart from the ill effects of social inequities on health, other stressors—such as poverty and violence—tax health and well-being at the individual level [44] by compromising the body’s ability to mobilize and demobilize physical mechanisms for adapting to stress [45]. The term “allostatic load” represents the failure of physiological homeostatic mechanisms to respond flexibly to chronic, unremitting stress [46]. High allostatic load is linked to serious physical and mental illnesses such as asthma [47], cancer [48], autoimmune diseases [49], and depression [50]. 

The social determinants of health and mental health literature calls for more equitable distribution of access to resources such as education, employment, health care, nutrition, and positive social supports [51,52]; we argue that support for animal companionship through HASS must be explicitly recognized as positive social support within public policies at the local, state, and federal level. Mechanistically, public policies shape access to resources, which impacts downstream, proximal risk factors directly related to health and well-being [51]. For example, health care policies that financially restrict access to services impair the ability of economically disadvantaged individuals to obtain timely treatment for health conditions. Un- or undertreated illness further limits an individual’s ability to generate income, continuing a positive feedback cycle that worsens health. At such times, individuals with companion animals may be at particular risk of crises leading to temporary or permanent separations from their animals; HASS can alleviate such risk. 

The OHOW model suggests extending health promotion efforts to include the physical and psychological symbiosis between humans and animals—the beneficial effects of this bond when both are well and the harms that emerge when either is ill. Many existing One Health models currently favor promotion of animal health primarily in service to human health. Rabies vaccination provides an exemplar of this zoonosis-based priority—in many countries, the justification for promoting animal health lies in preventing human disease. This one-sided approach fails to recognize the importance of well-being in the context of biodiversity—when one species does not do well, many others are inevitably impacted.

A practice example of an OHOW approach relates to human psychological assessments. Such assessments typically collect information about the social networks and supports that surround individuals being evaluated for mental health treatment. Typical questions ask about clients’ family members, friendships, spiritual practices, and other sources of support. Questions regarding companion animals are routinely missing from these standard evaluations, yet some patients regard their animals as their main reason for living [53]. An OHOW approach to mental health assessment would query clients about the role of companion animals in their lives. The answers to these questions carry implications for intervening in the lives of the human client and the animals in their lives.

## 5. Conclusions

Addressing barriers to establishment of integrated human and animal services is no small task. Obstacles include siloed human/animal services, human-centric healthcare policy priorities, societal views of animals as objects or property, funding, and infrastructure. A model such as OHOW, encompassing HASS, requires coordination between multiple human- and animal-serving sectors. Co-located human service models can serve as a template for such integration. For example, human primary care and mental health providers can be physically co-located. When co-location is infeasible, interdigitated referral pathways streamline providers’ ability to connect primary care patients with mental health services. Veterinary medicine has established similar practices, embedding veterinary social workers in emergency and specialty pet clinics or building relationships with local health and/or mental health service providers [36].

Funding and policy considerations represent a more difficult set of problems to solve. For example, human medical services are often funded by public or private insurers or federal programs established to bridge gaps in insurance coverage. Veterinary medicine is funded mostly by private payors or by pet insurance programs. Prioritizing human and animal health simultaneously necessitates complementary funding streams that facilitate access to both services. Putting into practice the Ottawa Charter approach of mediation is critical here; mediating and reconciling cross-sector differences in resources to maximize positive outcomes is essential for sustainable OHOW HASS practices. 

There are several examples of successful mediation across sectors within the United States that have resulted in public policy support of HASS. Federal examples of inter-agency collaboration supported by legislation and funding include the Pet and Women Safety Act for organizations that co-shelter survivors of intimate partner violence and their companion animals [54] and the Pet Evacuation and Transportation Standards Act, which requires states to plan for evacuation and rescue of companion pets alongside their families in order to receive federal funding for disaster relief plans [55]. California provides a state-level example with its efforts to reduce homelessness by breaking down barriers to housing for people with companion animals. The Pet Assistance and Support Program allocated USD 10 million in set-aside funding for homeless shelters that accept people and their companion animals [56]. The impetus for these legislative efforts to support HASS was human-centric—each program was designed to prevent human morbidity and mortality resulting from people’s unwillingness to abandon their companion animals—however, the end result saves both human and animal lives.

The foregoing examples illustrate that intervening to improve human and animal welfare simultaneously is indeed possible. Building upon these exemplars, decision-makers at federal, state, and local levels can replicate policies and programs that fit the unique human–animal service needs of their respective constituents. Discussions among representatives of macro-, organizational-, and individual-level constituents are needed to move OHOW efforts forward. Advocacy with multi-national organizations could provide the impetus for these conversations and existing local One Health Access practices can serve as templates. Although there is no easy path to establishing OHOW healthcare as a universal practice model, the tenets of the Ottawa Charter offer guidance; doing so could ultimately catalyze improvement in human and animal well-being at population levels.

## Figures and Tables

**Table 1 ijerph-20-05272-t001:** Human–animal support services in a tiered public health framework.

HASS	Human Tier	Animal Tier	Human Tier Notes	Animal Tier Notes
Better Cities for Pets—certification process for cities to identify and meet pet-friendly benchmarks Better Cities for Pets https://www.bettercitiesforpets.com/ (accessed on 12 March 2023)	Primary	Primary	Promotes increased access to/enjoyment of built and natural environments together (increased time engaged in HAB benefit activities)	Promotes increased access to/enjoyment of built and natural environments together (increased time engaged in HAB benefit activities)
Healthy Bond for Life—American Heart Association prevention/public health initiative https://www.heart.org/en/healthy-living/healthy-bond-for-life-pets (accessed on 12 March 2023)	Primary	Primary	Educates on protective and preventative aspects of benefits of HAB/human companionship and encourages engagement in related activities	Animals’ people may be motivated to increase positive/healthy activities with animals
Affordable pet adoption program Fee waived adoption research— MacArthur et al., 2012 [33]. Weiss et al., 2009 [34].	Primary Secondary	Tertiary	Primary—promotes well-being associated with animal companionship and HAB benefits Secondary—promotes well-being associated with animal companionship and HAB benefits for members of at-risk populations who choose to adopt	Homeless animal receives loving permanent home and human companionship benefits
Accessible veterinary care—wellness, prevention—located in animal care setting Smith et al., 2022 [35]	Primary	Primary	Maintains animal companionship/HAB benefits; prevented/reduces HAB stressors (finance, emotional) due to animal health issues; zoonotic transmission prevented	Health issues prevented; maintains human companionship/HAB benefits
Accessible veterinary care—treatment and wellness— located in animal care setting Smith et al., 2022 [35]	Primary Secondary	Primary Secondary Tertiary	Primary—maintains animal companionship/HAB benefits; zoonotic transmission prevented Secondary— preventes/reduces HAB stressors (finance, emotional) due to animal health issues	Primary—health issues prevented; maintains human companionship/HAB benefits Secondary—emerging health risks addressed; risk of losing home/person due to untreated health/ behavior problems reduced Tertiary—health/ behavior issues treated
One Health Clinic—integrated human primary care and veterinary wellness/treatment One Health Clinic—Tin et al., 2022 [36]	Primary Secondary Tertiary	Primary Secondary Tertiary	Primary—seeking care for animal can activate use of healthcare for self; health issues prevented; maintains animal companionship/HAB benefits; zoonotic transmission prevented Secondary— emerging health risks addressed; preventes/reduces HAB stressors (financial, emotional) due to animal health issues; Tertiary—health/ behavior issues treated	Primary—health issues prevented; maintains human companionship/HAB benefits; zoonotic transmission prevented Secondary—emerging health risks addressed; reduces risk of losing home/person due to untreated health/ behavior problems Tertiary—health/ behavior issues treated
Post-adoption animal training/behavior support program The Arizona Pet Project Post-adoption follow up calls and behavior support https://azpetproject.org/ (accessed on 12 March 2023)	Primary Secondary	Primary Secondary	Primary—increases positive communication and interactions with animal, can strengthen HAB Secondary—reduces animal-related stress and helps maintain animal companionship/HAB benefits	Primary—increases positive communication and interactions with animal, can strengthen HAB Secondary—reduces shelter surrender risk due to behavior issues
Pet deposit fee assistance program Arizona Humane Society Bridge the Gap program https://www.azhumane.org/resources-to-keep-your-pet/ (accessed on 12 March 2023)	Secondary	Secondary Tertiary	Secondary—can prevent homelessness among those at-risk; human undergoing stress keeps animal companion/HAB benefits; trauma of relationship rupture with animal avoided Tertiary—unhoused person is helped to afford housing with animal	Secondary—risk of shelter surrender reduced; trauma of relationship rupture with human prevented Tertiary—animal who is with unhoused person moves into affordable housing with person
Co-sheltering people and pets together Gillespie et al., 2017 [37]	Secondary Tertiary	Secondary Tertiary	Secondary—human undergoing stress has ongoing access to protective human companionship/HAB benefits Tertiary—person is safely sheltered; trauma of relationship rupture with animal prevented	Secondary—animal undergoing stress has ongoing access to protective human companionship/HAB benefits Tertiary—animal homelessness prevented; trauma of relationship rupture with human prevented
Pet food bank—located in animal welfare entity Schor et al., 2021 [38]	Secondary	Secondary Tertiary	Human undergoing stress keeps animal companionship/HAB benefits	Secondary—reduces risk shelter surrender risk due to inability to afford petfood Tertiary—provides food to alleviate hunger
Pet food bank—located in human food pantry Rauktis et al., 2017 [39]	Secondary Tertiary	Secondary Tertiary	Secondary—human undergoing stress keeps animal companionship/HAB benefits Tertiary—may motivate person to access food and alleviate own hunger due to need to seek pet food (Rauktis, 2017 [39])	Secondary—reduces shelter surrender risk due to inability to afford pet food Tertiary—provides food to alleviate hunger
Temporary pet foster care for people in crisis (economic, health, dv, etc.)—human and animal separated and reunified, e.g., Home Away From Home Program at Arizona Humane Society https://www.azhumane.org/resources-to-keep-your-pet/ (accessed on 12 March 2023)	Tertiary	Tertiary	Person is able to access health care, shelter, or other resource that precludes animal accompanying; trauma of permanent relationship rupture with animal prevented	Animal homelessness prevented; trauma of permanent relationship rupture with human prevented
Hope and Recovery Pet (HARP) Program—places shelter pets as ESAs for individuals with severe mental illness and social isolation Hoy-Gerlach et al., 2022 [40]	Tertiary	Tertiary	Loneliness and symptoms of mental illness can be reduced by living with animal companion/HAB benefits	Homeless animal receives loving permanent home, care, and human companionship

## Data Availability

Not applicable.

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
