# Peer review of "Reimagining Healthcare: Human–Animal Bond Support as a Primary, Secondary, and Tertiary Public Health Intervention"

_ijerph, 2023, doi:10.3390/ijerph20075272_

Round 1

Reviewer 1 Report

Overall, I really appreciated reading this article, I found it easy to read and well-written (I’m not a native English speaker though), and with a clear logical thread. The introduction is particularly straightforward, and it is very much appreciated to have a clear idea of where we are heading shortly after starting to read. On a side note, I found that sometimes the paper could be strengthened by references or a little bit more of explanations about the points being made.

- l.31-32 “the importance of the human-animal bond for mutual well-being is being increasingly recognized across human and animal health and welfare organizations across the globe.”, seems like a fact, could it be strengthened by a reference?

- I would strengthen a little bit the part about the health benefits associated to pet companionship with a little more of reference (e.g., Friedmann & Son 2009; Knight & Edwards, 2008; Nimer & Lundahl, 2007, but see Amiot & Bastian, 2015 for a review), because after all this is the core idea of HASS, right? Additionally, some dissenting voices begin to argue that this the relation between pets and health benefits is not so simple (see Mueller et al., 2021; Rodriguez et al., 2021), perhaps it might be interesting to raise that at one point or another in the paper? I leave to the authors evaluate the relevance of this suggestion.

- typo l.210 “conflacted”

- l.213-219 I think something is wrong in the format of this section. “Consequently, they have been combined… Health promotion focuses (…)” is that possible to make a sentence to get rid of the “…”. Plus, it seems that you are reporting a citation, but where does it start exactly? I failed to find the square brackets at the beginning of it.

- l. 238 & 258, I failed to find the meaning of the HAB acronym. Could author detail it, or maybe get rid of the acronym and present the full terms as it is only occurring 2 times. 

- formatting issue with Table 1, not the same police size between p.6, 7, and 8. Potentially due to the article-like formatting process.

- Some acronyms are created but never used, such as the Pet and Women Safety (PAWS), Pet Evacuation and Transportation Standards Act (PETS), or The Pet Assistance and Support Program (PAS), that are only mentioned one time. Authors might check whether it is relevant to create acronyms in this situation.

Author Response

Thank you for your feedback, very helpful!

Reviewer 2 Report

The work presented in this paper sounds interesting and it is based on the aspects of growing importance of the human-animal bond for mutual well-being and these issues are increasingly recognized across human and animal health and welfare organizations across the globe. The manuscript is a review that aims to recognize and explicate services that support human-companion animal relationships and to delineate examples of human-animal support services (HASS) within the three-tiered public health intervention framework.
General comments: This manuscript has focused on the 3 main aspects: understanding health: more than absence of disease, HASS: a one health approach and understanding HASS as public health interventions. These three issues seem to be the keys to recognize services supporting human-companion animal interactions as well as to delineate examples of human-animal support services (HASS) within the three-tiered public health intervention framework.
The work is generally good, although some minor revisions (in the Abstract, the Introduction section) can be done to improve the quality of the paper and attract a wider readership. Suggested changes are included in the specific comments.
Specific comments: This manuscript in my opinion is a kind of review paper. Considering the type of manuscript for now it is selected as article and not as review and it is suggested to improve the type of manuscript (to change it to review). According to previous comment also some improvements could be done in the Abstract section. As it is a kind of review paper it is not useful to divide abstract into the subsections such as background, methods, results and conclusion, so I recommend to remove these subsections’ names.
The introduction is generally fine but it is quite short and general so it could be improved. ‘Human-animal bond’ is present in the title of the manuscript as well as in its key words, so it is recommended to add some more information about this issue in the Introduction section. Moreover, it could be underlined that the issues in the paper are mainly connected with companion animals and these animals’ bond with humans.

Author Response

(The authors gave the same response as above.)
